# High-Risk Neuroblastoma: A Surgical Perspective

**DOI:** 10.3390/children10020388

**Published:** 2023-02-16

**Authors:** Jillian C. Jacobson, Rachael A. Clark, Dai H. Chung

**Affiliations:** 1Department of Pediatric Surgery, University of Texas Southwestern Medical Center, Dallas, TX 75390, USA; 2Children’s Health, Dallas, TX 75235, USA

**Keywords:** neuroblastoma, high-risk, pediatric surgical oncology, surgical resection

## Abstract

High-risk neuroblastoma requires multimodal treatment including systemic chemotherapy, surgical resection, radiation therapy, stem cell transplant, and immunotherapy. Surgeons play a vital role in obtaining local control of neuroblastoma and must therefore be knowledgeable about this complex pathology. This article provides a review of the optimal timing and extent of resection, the impact of various image-defined risk factors on surgical planning, and surgical approaches and techniques to enhance the resection of tumors in different anatomic locations.

## 1. Introduction

Neuroblastoma is the most common extracranial solid tumor in infants and children [1]. It accounts for 15% of pediatric cancer-related deaths [2]. Approximately 40% of newly diagnosed patients are defined as high-risk and many present with metastatic disease [3]. Although treatment advances in neuroblastoma now involve multi-modality therapy including chemotherapy, radiation, stem cell transplants, and immunotherapy, pediatric surgeons continue to play a vital role in obtaining local control of neuroblastoma.

### History

One of the earliest descriptions of neuroblastoma is thought to have been by Rudolf Virchow in the 1860s, who described this pediatric intra-abdominal mass as an “abdominal glioma” [4,5]. In the early 1900s, neuroblastoma was better characterized by William Pepper, Robert Hutchison, and James Homer Wright [4,6]. Pepper published a case series in 1901 about infants in Philadelphia who died after presenting with adrenal tumors and extensive hepatic infiltration without evidence of bony metastases [6,7]. In 1907, Hutchison reported a case series of pediatric patients in London who presented with adrenal tumors and bony metastases to the skull and orbits [8]. In 1910, after reviewing the findings of Hutchison and Pepper, Wright, a pathologist at Massachusetts General Hospital, published a case series of patients with adrenal tumors and disseminated liver involvement and/or bony metastases. Although Pepper and Hutchison had described these tumors as sarcomas, Wright identified microscopic “rosettes” of adrenal tumor tissue that resembled fetal adrenal tissue. He concluded that these adrenal tumors were of primitive neural origin and described them as “neuroblastomas” [6,9].

The role for surgery in the treatment of neuroblastoma was first described in the 1950s. In 1953, Robert E. Gross published “*The Surgery of Infancy and Childhood*”, explaining the role of extensive resection of neuroblastoma. He wrote how resection could lead to permanent cure, but that even when tumor must be left behind, tumor debulking improved the effectiveness of adjuvant radiation [10,11,12]. C. Everett Koop echoed these observations in his 1955 book, “*Neuroblastoma in Childhood: Survival After Major Surgical Insult to the Tumor*” [11,13]. Koop attempted to define risk status and described the role for surgery in resectable, nonresectable, and metastatic neuroblastomas [11].

The following review provides an overview of the current literature, advancements in pediatric surgical oncology, and the role of pediatric surgeons in the care of neuroblastoma patients today, including optimal timing, extent of resection, and preferred approaches utilized by surgeons for local control of this aggressive, heterogenous malignancy.

## 2. Timing of Surgical Resection

### 2.1. Techniques for Diagnosis

Although low-risk, early-stage tumors (International Neuroblastoma Staging System (INSS) stages 1, 2A, 2B) can often undergo upfront curative and complete resection, the initial surgical intervention of advanced and high-risk neuroblastomas is limited to biopsy [14].

Tissue diagnosis is required to confirm the diagnosis of neuroblastoma and to perform molecular studies to establish risk group status including: histology, grade of tumor differentiation, *MYCN* amplification status, and chromosomal aberrations of the tumor [3]. If indicated, central venous catheter/port placement and/or bone marrow aspiration can be performed during the same anesthetic event [11].

Traditionally, open incisional biopsy was the technique of choice for obtaining a tissue diagnosis. Typically, neuroblastomas are enclosed by pseudocapsules that can be opened to optimize post-operative hemostasis. Once the capsule is opened, multiple biopsies of at least 1 cm^3^ of tumor tissue should be obtained for analysis. More recently, minimally invasive, laparoscopic, and percutaneous core needle biopsy techniques have been utilized for tissue diagnosis. 

The previously described open technique can be adapted for a minimally invasive approach utilizing laparoscopic or thoracoscopic instruments for tumor visualization. Alternatively, a laparoscopic trocar can be placed to facilitate multiple needle passes for biopsies with subsequent packing of the needle tract with hemostatic agents [11].

In the last decade, multiple studies have compared the efficacy of percutaneous biopsy techniques to open incisional biopsy [15,16,17]. In 2014, Mullassery et al. published a retrospective single-center study of 39 patients who underwent biopsy for neuroblastoma from January 2002 to July 2013. Of these, 21 patients underwent open biopsy and 18 patients underwent core needle biopsy. All patients had adequate tissue for diagnosis and histologic and molecular analysis. No patients required repeat biopsy. There was one complication in the open group involving a patient who developed bleeding after biopsy of a stage 4 adrenal primary tumor that required blood transfusion and laparotomy [15]. 

Campagna et al. published a single-center, retrospective review of 34 patients who underwent either percutaneous or surgical (open or laparoscopic) biopsy at their institution between 2011 and 2015. Data obtained were also compared to previously published data from 2002–2010. The recent cohort (2011–2015) demonstrated increased utilization of ultrasound-guided percutaneous core needle biopsy, an increase in the number of core needle samples, and a decrease in incidence of complications, relative to the historic cohort (2002–2010). Adequacy of biopsy was not significantly different between the percutaneous and surgical biopsy groups in either cohort. The authors also found that larger tumors were more likely to have been biopsied percutaneously [16]. 

The Pediatric Surgical Oncology Research Collaborative demonstrated similar results in their 2020 multi-institutional, retrospective study of 243 patients. There was no significant difference in the ability to obtain a primary diagnosis or determine *MYCN* amplification status between the open incisional biopsy and percutaneous core needle biopsy (PCNB) groups. Complications did not differ between groups, although the PCNB group did require fewer blood transfusions and demonstrated lower opioid usage [17]. Together, these findings suggested that image-guided percutaneous needle biopsy techniques were a safe and effective means for obtaining tissue diagnosis and noninferior to surgical biopsy techniques.

### 2.2. Optimal Timing of Surgical Resection

Neuroblastoma is a chemosensitive cancer that is often bulky and encasing vital vascular structures at presentation [18]. Many patients with low-risk neuroblastoma can undergo upfront resection [4,19]. Patients with intermediate-risk neuroblastoma typically benefit from neoadjuvant chemotherapy prior to consideration for surgical resection [19,20]. Current protocols for high-risk neuroblastoma treatment, however, involve three phases: induction, consolidation, and post-consolidation/maintenance. The induction phase includes multiagent chemotherapy, surgical resection (typically after four to six rounds of chemotherapy or at the completion of induction), and stem cell harvest. The consolidation phase involves high-dose chemotherapy and autologous hematopoietic stem cell rescue, followed by radiation. The post-consolidation phase typically includes immunotherapy (i.e., anti-disialoganglioside (GD2) antibody), cytokines, and isotretinoin [18,21,22,23,24,25].

Surgery and radiation therapy are the two mainstays of treatment for local control in neuroblastoma. However, they are two components of a multimodal treatment plan that, in the case of high-risk neuroblastoma, typically occur over approximately 18 months [18]. The current standard of care is for surgery to be performed during the induction phase after neoadjuvant chemotherapy in order to improve the resectability of the tumor. The Children’s Cancer Group study CCG3891 demonstrated improved resectability of primary tumors after neoadjuvant chemotherapy, with higher rates of complete resection or <5% residual tumor relative to patients who underwent resection at time of diagnosis [18,26]. There remains debate, however, as to the optimal time of resection. 

Proponents of earlier resection quote the current literature, including a study by Medary et al. demonstrating that the majority of tumor regression occurs within the first two cycles of chemotherapy, with minimal reduction in subsequent cycles [27]. Furthermore, additional neoadjuvant chemotherapy can contribute to fibrosis that may increase the difficulty of dissection [18]. Recent Children’s Oncology Group (COG) protocols advocate for surgery at the time of greatest tumor size reduction prior to the onset of significant fibrosis [25]. 

International Society of Paediatric Oncology Europe Neuroblastoma Group (SIOPEN) protocols, however, advocate for resection at the completion of induction to facilitate evaluation of local and metastatic disease response [24]. Furthermore, resection was not performed during induction in order to maintain their 10-day chemotherapy schedule and to achieve proper dose intensity [24]. In addition, advocates of later resection also postulate that delaying surgery until after autologous hematopoietic stem cell rescue may prevent patients from undergoing the nephrotoxicity of myeloablative conditioning with a single kidney should they require nephrectomy at the time of surgery [18]. 

Although the majority of current protocols advocate for neoadjuvant chemotherapy prior to surgical resection, the optimal timing for surgery needs further elucidation and is currently dictated by clinical trial protocols.

### 2.3. Role of Neoadjuvant Chemotherapy in Reducing IDRFs

The International Neuroblastoma Risk Group (INRG) Staging System (INRGSS) was designed to risk-stratify neuroblastoma patients at the time of diagnosis prior to initiating treatment. It is based on the extent of the disease and the presence or absence of image-defined risk factors (IDRFs) (Appendix A: Table A1 and Table A2) [28]. INRG stage, in addition to patient age, *MYCN* amplification status, and tumor biology, dictate risk status [29]. 

Very low-, low-, intermediate-, and high-risk categories were initially stratified based on event-free survival (EFS) statistics seen in these various patient groups to allow physicians to better prognosticate patient disease. In the initial staging system, very low-, low-, intermediate-, and high-risk patients had greater than 85%, 76–85%, 50–75%, and <50% five-year EFS, respectively [3]. This system was revised in 2021 (Children’s Oncology Group neuroblastoma risk classifier version 2) to include segmental chromosome aberrations as an additional genomic biomarker [29]. For version 2, 5-year EFS and OS were: low-risk: 90.7% ± 1.1% and 97.9% ± 0.5%; intermediate-risk: 85.1% ± 1.4% and 95.8% ± 0.8%; and high-risk: 51.2% ± 1.4% and 62.5% ± 1.3%, respectively. The presence of IDRFs, stage, and risk status are vital considerations in patient treatment, including surgical resection. Most high-risk tumors are associated with at least one IDRF [28]. For this reason, numerous studies have been published evaluating the role of IDRFs in surgical resection and the potential benefit of chemotherapy in reducing IDRFs prior to resection.

Phelps et al. evaluated the impact of IDRFs on surgical resection and oncologic outcome in neuroblastoma patients treated at their institution between 2002 and 2017 [30]. Of 106 patients, 61% of patients had IDRFs at diagnosis. They found that *MYCN*-amplified and undifferentiated neuroblastomas had more IDRFs than non-*MYCN*-amplified or more differentiated tumors. In addition, they found that neuroblastomas resected after neoadjuvant chemotherapy were smaller than at diagnosis. Despite this decrease in size, the presence of IDRFs at time of resection was associated with increased intraoperative estimated blood loss (EBL), longer operative times, increased risk of adjacent organ resection, and increased incidence of intraoperative complications, especially vascular injuries. They also found an association between the presence of IDRFs at time of resection and an increase in length of stay (LOS) and post-operative intensive care unit (ICU) admissions. This suggests that the presence of IDRFs at time of resection is associated with increased operative difficulty. Despite this increased difficulty, Phelps et al. also demonstrated that IDRF-negative tumors did not have significantly different rates of gross total resection (GTR), five-year relapse-free survival, or overall survival (OS) relative to IDRF-positive tumors [30].

A 2020 meta-analysis by Parhar et al. of 19 retrospective cohort studies representing 1132 patients evaluated the association between IDRF status and rate of incomplete surgical resection, rate of post-operative complications, 5-year EFS, and 5-year OS. They found that IDRF-positive patients were less likely to undergo a GTR. The pooled risk ratios (RR) for incomplete resection in IDRF-positive patients compared to IDRF-negative patients were 2.45 in all studies and 2.64 in studies where IDRF status was determined pre-treatment. They also reported an increased surgical complication rate in the IDRF-positive group, with pooled RRs of 2.30 in all studies and 2.83 in pre-treatment IDRF studies, relative to the IDRF-negative group. The authors concluded that IDRF-positive patients demonstrated higher risk of 5-year relapse and mortality (pooled hazard ratios (HR) for EFS and OS were 2.08 and 2.44, respectively) [31].

Utilizing a retrospective review of 88 patients over 20 years and four high-risk neuroblastoma protocols, Mansfield et al. evaluated the effects of neoadjuvant therapy on IDRFs and the ability to achieve a GTR. They found that neoadjuvant chemotherapy reduced the number of IDRFs by approximately 2.9 ± 2.5 per patient and contributed to a decrease in tumor volume by 89.8% ± 18.9%. They concluded that patients with three or less IDRFs at time of resection had the highest odds ratio for >90% GTR. Regarding types of IDRFs, they found that invasion of the renal pedicle was both the most common IDRF and the IDRF least likely to resolve with neoadjuvant chemotherapy. The ability to achieve >90% GTR, however, was not different between patients with renal hilar involvement and those without. An overall elevated number of abdominal vascular IDRFs, especially those involving porta hepatis infiltration and/or celiac axis encasement, however, was associated with decreased ability to achieve >90% GTR [32]. Their findings support those of Phelps et al. and demonstrate a positive association between number of IDRFs and EBL, blood transfusion requirements, and LOS [30,32]. However, Mansfield et al. also concluded that patients with IDRFs had a longer delay to resuming chemotherapy [32].

The presence of IDRFs increases the difficulty of dissection and can be associated with a decreased rate of GTR and increased risk of surgical complications. IDRFs, however, do not typically prohibit surgical resection. Furthermore, despite the variety of surgical complications that can be associated with resection of neuroblastomas, and the increase in frequency of complications seen with increase in IDRFs, operative mortality is rare, even for high-risk cases [24,25]. Neoadjuvant chemotherapy is valuable in decreasing the number of IDRFs and tumor volume prior to surgical resection.

## 3. Optimal Extent of Surgical Resection

It is commonly accepted that the goal of surgical resection in neuroblastoma is a gross macroscopic resection while preserving vital organs and maintaining organ function, including maintenance of renal and neurologic function, and avoiding complications that may delay resumption of chemotherapy [18]. 

The goal of surgical management for low-risk neuroblastoma is complete resection of the primary tumor. Many patients have an overall survival approaching or exceeding 90% with surgical resection alone, particularly in the setting of asymptomatic, localized disease with favorable histology [4,19,33,34,35,36]. In fact, asymptomatic infants with small adrenal tumors and those with stage MS disease without hepatomegaly can typically be offered a trial of observation alone [37,38]. Chemotherapy is typically reserved for: patients whose tumors cannot be resected without unacceptable morbidity (i.e., unresectability due to encapsulation of vital structures); patients who undergo a less than 50% resection of their primary tumor or who demonstrate progressive disease; and those presenting with life-threatening symptoms or neurologic compromise [19]. However, these patients tend to demonstrate favorable outcomes even with less intense chemotherapy [19]. Disease recurrence is typically rare, and when it does occur, it is typically a locoregional recurrence that can be salvaged with surgery and/or chemotherapy [19]. 

The optimal extent of surgical resection in intermediate-risk disease remains controversial [19]. This may be due to the substantial heterogeneity of tumors included in this risk group. Patients with intermediate-risk neuroblastoma typically undergo biopsy followed by upfront chemotherapy to decrease tumor size, mitigate IDRFs and difficulty of resection, and/or to eliminate the need for resection [20]. In COG trial ANBL0531 (*n* = 404 patients with non-*MYCN*-amplified tumors), patients were treated with two, four, or eight cycles of upfront chemotherapy with or without surgery [20]. Treatment group was assigned based on prognostic factors, including unfavorable histology, diploidy, or segmental chromosomal aberrations [20]. Ultimate duration of therapy was determined by treatment response [20]. Most patients demonstrated a partial response (PR, 50–90% reduction in the primary tumor) or better and therefore did not undergo surgical resection [20]. Surgical resection was utilized in patients to achieve the desired therapy goal without incurring substantial risk to vital structures, whereas additional chemotherapy was provided to patients who did not meet the therapy goal [20]. ANBL0531 demonstrated that a reduction in therapy was appropriate for many patients with intermediate-risk disease, particularly those with localized disease, using an end-of-therapy goal of PR, as opposed to very good partial response (VGPR, greater than 90% reduction in the primary tumor) [20].

A resection with negative microscopic margins, i.e., an R0 resection, is typically not achievable with advanced stage or high-risk disease [11,18]. This goal is a departure from the surgical management of many other pediatric solid tumors, including sarcomas. Many pediatric surgical oncologists have therefore questioned the optimal extent of resection for high-risk neuroblastoma. This has prompted several recent studies evaluating the relationship between the extent of surgical resection and local progression, EFS, and OS.

A 2013 German prospective clinical trial, NB97, evaluated 278 patients, age 18 months or older, with high-risk, stage 4 neuroblastoma. Of the patients who underwent surgery after induction chemotherapy, 54.7% of patients underwent GTR and 30.6% of patients underwent an incomplete resection. The extent of resection did not impact local progression, 5-year EFS, or 5-year OS [39].

One of the largest retrospective multicenter studies, conducted by Fischer et al., included 179 patients, age >18 months with localized INSS stage 1–3 (including very low-, low-, intermediate-, and high-risk) neuroblastoma. They found that EFS correlated with the extent of resection: >95% resection was associated with 87.8% EFS, 90–95% resection was associated with 78.6% EFS, and 50–90% resection was associated with 66.7% EFS [40].

The COG A3973 prospective study of 220 neuroblastoma patients demonstrated similar findings [25]. In this study, 85% of patients had stage 4 disease, 1% had stage 4S disease, and 14% demonstrated either stage 2 or 3 disease. Surgeon-assessed extent of resection was ≥90% in 70% of patients (*n* = 154) and <90% in 30% of patients (*n* = 66). Patients who underwent ≥90% surgeon-assessed extent of resection demonstrated an improved EFS and decreased cumulative incidence of local progression (CILP) relative to patients in the <90% resection group (45.9% ± 4.3% EFS vs. 37.9% ± 7.2% EFS (*p* = 0.04), and 8.5% ± 2.3% CILP vs. 19.8% ± 5.0% CILP (*p* = 0.01), respectively). There was no significant difference in OS between groups. Interestingly, these findings were best apparent in patients with locoregional disease, as opposed to metastatic disease. This suggests that metastatic disease and its treatment response may play a greater role in prognostication than extent of local control. Major complication rates were not significantly different between groups; therefore, the authors concluded that ≥90% resection is feasible and safe in most patients with high-risk neuroblastoma. However, this study is limited by the subjective nature of surgeon-assessed extent of resection. The authors also utilized pre- and post-operative computed tomography (CT) scans to evaluate the concordance between surgeon assessment of extent of resection with cross-sectional imaging in a subset of patients. Concordance was low with an agreement of 63%. The explanation behind this discordance remains to be elucidated [25].

The prospective SIOPEN HR-NBL1 study also investigated the impact of surgeon-assessed extent of resection on local progression and survival in 1531 patients during the pre- and post-immunotherapy eras (2002–2015). Eleven hundred seventy-two patients (77%) underwent complete macroscopic excision (CME) and 359 (23%) underwent incomplete macroscopic excision (IME). CME was defined as removal of all visible and palpable tumor, as well as all involved lymph nodes. If visible or palpable tumor or involved lymph nodes remained after resection, the procedure qualified as IME, regardless of the volume of remaining tumor. Overall, there was a 0.46% mortality rate and a 9.7% rate of severe operative complications. Nephrectomy was performed in 8.8% of patients, typically in the setting of tumor encasement of the renal hilum, and was less frequent in the CME group. Overall, the SIOPEN HR-NBL1 study found that CILP, 5-year EFS, and 5-year OS were improved in the CME group relative to the IME group, both before and after the introduction of immunotherapy with dinutuximab beta. In addition, the SIOPEN group concluded that there is a survival benefit and reduction in CILP associated with CME plus radiotherapy. These superior outcomes are seen in both the pre- and post-immunotherapy eras. Given the low surgical complication and mortality rates, the SIOPEN group advocates for CME of primary tumors in patients with stage 4 high-risk neuroblastoma who responded to induction therapy [24].

Although the NB97 study did not find an association between extent of resection and CILP, EFS, and OS, subsequent retrospective and prospective studies suggest an improvement in CILP and EFS with CME or >90% tumor resection [24,25,39,40]. 

Further work is needed to standardize reporting of extent of resection in order to better determine its effect on outcomes. The varying locations of primary neuroblastic tumors and their tendency to encase vital structures can increase the difficulty of dissection and therefore the risk of morbidity and mortality to the patient. Surgeons must often weigh the impact of a more extensive resection on patient morbidity and overall oncologic outcomes. However, lack of standardization in reporting can be a limitation in effective, generalizable data analysis. A joint initiative by the Pediatric Oncological Cooperative Groups SIOPEN, COG, and Gesellschaft fuer Paediatrische Onkologie und Haematologie (GPOH)—German Association of Pediatric Oncology and Hematology, recently created a novel standard form for systematic reporting known as The International Neuroblastoma Surgical Report Form (INSRF) [41]. It consists of five sections to be completed by the surgical team: (1) patient and treatment protocol details; (2) type, timing, and date of intervention, extent of resection, localization of the primary tumor and details regarding the preoperative plan discussed at a multidisciplinary tumor board, plus a description of pre-chemotherapy IDRFs; (3) surgical findings (including structures/organs involved by tumor) and corresponding intraoperative management; (4) intraoperative complications and corresponding management; and (5) post-operative complications within 30 days of surgical intervention [41]. The second section includes four options for extent of resection: (1) “complete resection”, (2) “minimal residue”, defined as “less than 5 cubic centimeter of tumor remaining”, (3) “incomplete resection”, defined as “5 or more cubic centimeter residue”, and (4) “other” [41]. Standardized reporting of extent of resection, in conjunction with reported outcomes, will likely allow for more robust analysis regarding the extent of resection on patient morbidity and overall outcomes in neuroblastoma.

## 4. Surgical Approaches and Techniques by Anatomic Location

The optimal pre-operative evaluation and surgical approach for neuroblastoma resection varies by anatomic location. Important anatomic considerations specific to each location, including exposure and possible complications, are summarized below.

### 4.1. Cervical and Cervicothoracic Lesions

Primary cervical lesions typically occur in patients under one year of age and possess favorable biologic and prognostic features [4,11]. The presentation of these lesions can vary. Possible presentations include: a congenital or palpable neck mass, Horner’s syndrome or anisocoria secondary to cranial nerve or sympathetic chain compression, snoring or respiratory distress secondary to upper airway compression, and dysphagia or aspiration due to compression of the digestive tract [42]. 

Surgical planning of cervical lesions includes determination of the extent of the tumor and assessment for involvement or invasion of the thoracic inlet. A transverse neck incision can be used for carotid sheath exploration [4,11]. Larger tumors may require division of the strap muscles, including the sternocleidomastoid, for adequate exposure. Lymph node or neck dissection is indicated if lymph nodes are grossly positive for malignancy. Potential post-operative complications include Horner syndrome, characterized by ipsilateral ptosis, pupillary miosis, and facial anhidrosis [4,11].

Tumors invading the thoracic inlet typically necessitate trap-door thoracotomy for adequate exposure. The patient’s ipsilateral arm is placed in 90-degree abduction. An incision is made superior and parallel to the ipsilateral clavicle, down the midline overlying the sternum to the fifth intercostal space, and then laterally, stopping at the ipsilateral anterior axillary line. The sternum is divided and a retractor is placed between its cut edges. Tumors with larger cervical components may require extension of the neck incision along the anterior border of the ipsilateral sternocleidomastoid muscle. Tumors involving bilateral thoraces can be accessed using a clamshell thoracotomy. Potential post-operative complications of cervicothoracic tumor resection include nerve injury (e.g., vagus nerve, long thoracic, phrenic nerve, or brachial plexus), pneumothorax, or vascular injury to any structures the tumor may be encasing or invading [4,11].

### 4.2. Mediastinal Lesions and Thoracoabdominal Lesions

The posterior mediastinum is a very common primary site of neuroblastoma, second only to the adrenal gland and upper abdominal sympathetic ganglia. Adequate exposure can typically be obtained through a muscle-sparing posterolateral thoracotomy or video-assisted thoracoscopy (VATS) [4,11]. Tumors invading the spinal foramina may necessitate foraminotomy. Much like cervical tumors, Horner syndrome is a possible complication of resection of superior mediastinal lesions due to tumor proximity to the stellate ganglion [4,11]. Chylothorax is also a possible complication of resection of mediastinal lesions given the proximity to the thoracic duct.

Resection of inferior, posterior mediastinal tumors and certain thoracoabdominal tumors carries a risk of paraparesis or paralysis secondary to spinal cord ischemia due to tumor proximity to the artery of Adamkiewicz (AoA). The AoA is an intraforaminal artery that typically arises between T9 and T12 from a left-sided intercostal or lumbar artery. It then courses anteriorly and cephalad along the spinal cord before joining the anterior spinal artery in a hairpin curve [43,44,45]. The approximately 0.92–1.89 mm caliber artery follows a relatively characteristic course; however, variability is not uncommon [46]. Spinal angiography is the radiographic gold standard for identification of the AoA [47,48]. Recent case reports of post-operative paraplegia secondary to spinal ischemia after iatrogenic injury to the AoA have prompted pediatric surgeons to advocate for the utilization of pre-operative spinal angiography. Angiography allows for identification of the location of this artery and evaluation of its relationship to the lesion to determine resectability [49,50,51]. 

Exposure of transdiaphragmatic thoracoabdominal neuroblastomas presents a unique challenge to surgeons. Martucciello et al. described a thoracophrenolaparotomic (TPL) approach for enhancing exposure [50]. Patients were positioned in the ipsilateral oblique lateral decubitus position with their arm bent over their head. An incision was made along the course of the tenth rib and then down along the lateral margin of the rectus abdominis muscle beyond the umbilical transverse line. A tenth rib subperichondral and periosteal-sparing resection was performed to allow for post-operative rib regrowth and the pleura was entered. A laparotomy was performed along the lateral margin of the rectus. The diaphragm was incised radially along the posterior peripheral margin to complete exposure without injuring the phrenic nerve. Radical tumor resection was performed. The diaphragm was repaired, and a chest tube and retroperitoneal drain were secured [50].

Thoracoabdominal and two-cavity approaches have also been described for resection of thoracoabdominal neuroblastomas. When performing a thoracoabdominal approach, the patient is typically placed supine with a bump underneath the side of the body from which the tumor originates. As described by Qureshi and Patil, an incision is made beginning superior to the umbilicus and extended obliquely over the abdomen across the costal margin and the eighth intercostal space to the posterior axillary line below the inferior angle of the scapula [52]. The muscles are divided with electrocautery. The diaphragm is typically divided radially, as opposed to circumferentially, taking care to avoid injury to the phrenic nerve [52]. If further exposure is needed, the thoracoabdominal incision can be combined with a midline abdominal incision extending cephalad or caudad [52]. This approach has been utilized and modified. Described variations include a transverse laparotomy with extension perpendicularly to the thorax through the mid-axillary line, again taking great care to avoid injury to the phrenic nerve [53]. Two-cavity approaches typically consist of a laparotomy plus either thoracotomy, thoracoscopy, or incision of the diaphragm [53]. Regardless of the approach, improving exposure aids in mitigating injury and enhancing oncologic resection. Pain control should be optimized post-operatively to aid with patient recovery.

### 4.3. Upper Abdominal and Retroperitoneal Lesions

Most neuroblastomas originate in the adrenal gland or sympathetic ganglia in the upper abdomen. Many neuroblastomas are diagnosed at more advanced stages. Therefore, involvement of regional para-aortic, pericaval, and interaortocaval lymph node chains is common. Encasement of the surrounding vasculature is also frequently seen in high-risk neuroblastomas. A thoracoabdominal approach can be utilized to improve exposure of these vessels [4,11]. Exposure of the vena cava or right hilar vessels is best obtained by a right-sided incision, while a left-sided incision is more optimal for tumors encasing the aorta, celiac axis, superior mesenteric artery, or left hilar vessels [11]. Due to the involvement of vascular structures in this region, vascular injury is a more frequent complication of abdominal neuroblastoma resections. 

As previously stated, invasion of the renal pedicle by neuroblastoma tumors is the most common IDRF and the least likely IDRF to resolve with chemotherapy [32]. Therefore, although the kidney should be preserved whenever possible, there is always a risk of partial or total nephrectomy, as well as renal infarction, with any surgical attempt at locoregional control of tumors with renal involvement. Injury to or loss of a kidney has been described in up to 15% of cases [54]. Fahy et al. retrospectively analyzed long-term outcomes of patients who underwent nephrectomy during surgical resection of high-risk, intra-abdominal neuroblastomas. They found that children who underwent nephrectomy, as opposed to kidney-sparing surgery, did not demonstrate any statistically significant difference in recurrence or OS. However, estimated GFR was decreased in the nephrectomy group (90 mL/min/1.73 m^2^ nephrectomy group vs. 127 mL/min/1.73 m^2^ kidney-sparing group) [55].

### 4.4. Pelvic Lesions

Although primary pelvic neuroblastomas typically possess favorable prognostic factors and biology without distant metastasis, resection can be complicated by encasement of iliac vessels and tumor proximity to the lumbosacral plexus. An infraumbilical, low vertical midline incision can be utilized to obtain exposure in this region, allowing for proximal control of the distal aorta and inferior vena cava with optimal exposure of the iliac vessels. Internal iliac vessels can be ligated and divided if needed for GTR of pelvic neuroblastomas. Possible post-operative complications of pelvic tumor resections include vascular injury and foot drop secondary to injury of pelvic nerve roots [11].

### 4.5. Is There a Role for MIS Resections of Neuroblastoma?

While open resection remains the standard of care, there are a growing number of studies evaluating the role of minimally invasive surgery (MIS) in pediatric oncologic resections. Traditionally, MIS has been associated with less post-operative pain, decreased risk of post-operative ileus and adhesions, decreased time to feeding, and improved cosmesis, compared to open surgery, making it an attractive alternative to open operations when safe and feasible [56,57,58].

A retrospective study by Kelleher et al. compared clinical outcomes in pediatric adrenal neuroblastoma patients undergoing laparoscopic versus open adrenalectomy. Criteria for laparoscopic resection included the absence of vascular encasement and a maximum tumor size of five centimeters in greatest dimension at the time of surgery. There was no significant difference in recurrence or survival between the open and laparoscopic groups regardless of risk stratification. Furthermore, patients who underwent laparoscopic resection demonstrated shorter LOS relative to the open group regardless of risk category [59].

Shirota et al. compared outcomes of laparotomy vs. laparoscopy for neuroblastoma resections, focusing on patients without IDRFs. The authors found no significant difference in operative time, 1-year locoregional recurrence, progression-free survival, or OS rates between groups. The laparotomy group, however, demonstrated higher EBL and increased time to resumption of oral intake [60]. 

Phelps et al. evaluated laparoscopic vs. open resection of pediatric embryonal tumors, of which, the majority of MIS resections were for neuroblastic tumors (*n* = 20, 83%). The authors found that 5-year relapse-free and overall survival were not significantly different between groups for resection of tumors with volumes less than 100 mL. There was also no difference in margin status or lymph node sampling between MIS and open groups. Furthermore, the MIS group demonstrated decreased EBL, operative times, and LOS. However, most patients who underwent MIS resections were notably of older age with larger body surface area and they commonly had earlier stage, lower risk tumors with fewer or zero IDRFs, demonstrating the importance of patient selection on surgical approach [61].

Similar findings have been demonstrated with resection of thoracic neuroblastomas. Malek et al. compared thoracoscopic vs. open resections in their single-institution, retrospective chart review of neuroblastoma patients between 1990 and 2007. Out of 37 patients, 26 patients underwent thoracotomy and 11 patients underwent thoracoscopy. There were no statistically significant differences regarding patient demographics or tumor characteristics between groups. In addition, there were no statistically significant differences regarding operative time, complications, recurrence, disease-free survival, or OS between groups. The thoracoscopic group, however, had decreased EBL and LOS relative to the open group [62].

Overall, MIS has been shown to be associated with decreased EBL and LOS without negative effects on incidence of operative complications, recurrence, or survival in resection of abdominal and thoracic neuroblastomas. Although randomized controlled trials are needed to better determine the role for and effectiveness of MIS resections, current data supports consideration of minimally invasive resection techniques for smaller tumors without vascular encasement [63].

## 5. Conclusions

Despite continued research and surgical innovation, high-risk neuroblastoma remains difficult to treat with less than 50% disease-free survival at five years [1]. Surgery is a vital component in the diagnosis and multimodal treatment algorithm for high-risk neuroblastoma, and along with radiation therapy, is necessary for achieving local control. Complete macroscopic excision can be curative in localized or low-intermediate risk neuroblastoma and contributes to a reduction in local progression and an increase in EFS in high-risk neuroblastoma. Current data suggests that although local control is important, poor metastatic response may explain the similar OS among patients with different extents of surgical resection. Therefore, further research is needed to improve survival in patients with advanced or metastatic disease. In addition, randomized controlled trials and clinical trials are required to better determine the role for minimally invasive surgery in the treatment of neuroblastoma.

## Data Availability

No new data were created or analyzed in this study. Data sharing is not applicable to this article.

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
