# Peer review of "High-Risk Neuroblastoma: A Surgical Perspective"

_children, 2023, doi:10.3390/children10020388_

Round 1

Reviewer 1 Report

This manuscript is review about surgical intervention for high-risk neuroblastoma. This report has some concerns.

In Line 110(Page3) Please spell 'GD2' correctly.

In Line 140(Page3) Please describe INGRSS using Figure or Table.

Please quote 'A novel standard for systematic reporting of neuroblastoma surgery' Annals of Surgery  doi: 10.1097/SLA.0000000000003947. associating with the definition of resection.

Author Response

This manuscript is review about surgical intervention for high-risk neuroblastoma. This report has some concerns.

In Line 110(Page3) Please spell 'GD2' correctly.

           The sentence now reads “(i.e., anti-disialoganglioside (GD2) antibody).”

In Line 140(Page3) Please describe INGRSS using Figure or Table.

           We appreciate the reviewer’s comment. The INRGSS is now described in Table 1.

Please quote 'A novel standard for systematic reporting of neuroblastoma surgery' Annals of Surgery  doi: 10.1097/SLA.0000000000003947. associating with the definition of resection.

           We thank the reviewer for this comment. The last paragraph of the “Optimal Extent of Surgical Resection” now discusses the paper, “A Novel Standard for Systematic Reporting of Neuroblastoma Surgery: The International Neuroblastoma Surgical Report Form,” including its definition of resection. This new paragraph is copied below:

“Further work is needed to standardize reporting of extent of resection in order to better determine its effect on outcomes. The varying locations of primary neuroblastic tumors and their tendency to encase vital structures can increase the difficulty of dissection and therefore the risk of morbidity and mortality to the patient. Surgeons must often weigh the impact of a more extensive resection on patient morbidity and overall oncologic outcomes. However, lack of standardization in reporting can be a limitation in effective, generalizable data analysis. A joint initiative by the Pediatric Oncological Cooperative Groups SIOPEN, COG, and Gesellschaft fuer Paediatrische Onkologie und Haematologie (GPOH) – German Association of Pediatric Oncology and Haematology, recently created a novel standard form for systematic reporting known as The International Neuroblastoma Surgical Report Form (INSRF).1 It consists of five sections to be completed by the surgical team: (1) patient and treatment protocol details, (2) type, timing, and date of intervention, extent of resection, localization of the primary tumor and details regarding the preoperative plan discussed at a multidisciplinary tumor board, as well as a description of pre-chemotherapy IDRFs, (3) surgical findings (including structures/organs involved by tumor) and corresponding intraoperative management, (4) intraoperative complications and corresponding management, and (5) postoperative complications within 30 days of surgical intervention.1 The second section includes four options for extent of resection: (1) “complete resection,” (2) “minimal residue,” defined as “less than 5 cubic centimeter of tumor remaining,” (3) “incomplete resection,” defined as “5 or more cubic centimeter residue,” and (4) “other.”1 Standardized reporting of extent of resection, in conjunction with reported outcomes, will likely allow for more robust analysis regarding the extent of resection on patient morbidity and overall outcomes in neuroblastoma.”

Reviewer 2 Report

This paper describes a review of a surgical aspect of high-risk neuroblastoma. The manuscript is educational for pediatricians and pediatric surgeons, but I ask the authors to address the following comments.

1.      Page 2. Line 11 & 82. INSS stage is written in Arabic numerals in linw11 while the other is written in alphabetical characters in line 82. Please unify them.

2.      Page 6, line 283. Finochietto” should be removed. 

3.      Page 7, line 314-324. This sentence is redundant. Instead of introducing the technique in the paper, it would be better to write about how it is more effective than other techniques and what other techniques are available for TPL.

4.      Page 8, line 377. This paper evaluates the efficacy of MIS in embryonal tumors, but it seems to have a significant selection bias, and the authors write as such. Especially in neuroblastoma, IDRF is mentioned and not only in volume of tumor, which may need to be written. Or maybe this paragraph is unnecessary.

Author Response

1. Page 2. Line 11 & 82. INSS stage is written in Arabic numerals in linw11 while the other is written in alphabetical characters in line 82. Please unify them.

We thank the reviewer for this comment and all stages are now written with Arabic numerals for consistency.

2. Page 6, line 283. “Finochietto” should be removed.

The word “Finochietto” has been removed.

3. Page 7, line 314-324. This sentence is redundant. Instead of introducing the technique in the paper, it would be better to write about how it is more effective than other techniques and what other techniques are available for TPL.

We thank the reviewer for this thoughtful comment. We have deleted several sentences within this paragraph to reduce redundancy. However, we feel that a cursory description of this technique is valuable in a paper describing surgical management of neuroblastoma in order to hopefully enhance readers’ knowledge of these advantageous surgical techniques when caring for children with these challenging tumors. Furthermore, we have also added information about additional techniques for thoracoabdominal tumors, as the reviewer suggested.

4. Page 8, line 377. This paper evaluates the efficacy of MIS in embryonal tumors, but it seems to have a significant selection bias, and the authors write as such. Especially in neuroblastoma, IDRF is mentioned and not only in volume of tumor, which may need to be written. Or maybe this paragraph is unnecessary.

We thank the reviewer for this insightful comment. We agree with the reviewer’s concerns and have elaborated on these details in our discussion of this paper. This paragraph now reads: “Phelps et al. evaluated laparoscopic vs. open resection of pediatric embryonal tumors, of which, the majority of MIS resections were for neuroblastic tumors (n=20, 83%). The authors found that 5-year relapse-free and overall survival were not significantly different between groups for resection of tumors with volumes less than 100 mL. There was also no difference in margin status or lymph node sampling between MIS and open groups. Furthermore, the MIS group demonstrated decreased EBL, operative times, and LOS. However, most patients who underwent MIS resections were notably of older age with larger body surface area and they commonly had earlier stage, lower risk tumors with fewer or zero IDRFs, demonstrating the importance of patient selection on surgical approach.

Reviewer 3 Report

Well written paper, description of the different options availables in obtaining diagnosis and resection of NB is adequately complete. The paper is fluid and can be a good starting point from a didactical point of view. A politically correct light review.

Author Response

Well written paper, description of the different options availables in obtaining diagnosis and resection of NB is adequately complete. The paper is fluid and can be a good starting point from a didactical point of view. A politically correct light review.

We thank the reviewer for these kind comments.

Reviewer 4 Report

This is a very nice review of the surgical considerations in neuroblastoma. The great majority of the paper seems to be dedicated to high risk disease, though not exclusively.

In some areas, such as the techniques for diagnosis, there is sufficient differentiation between non-HR and HR. In other areas, most prominently the section describing the studies looking at extent of resection, the discussion is very focused on HR disease without much acknowledgment of the lower risk tumors. It would be helpful to include some information about resection approaches in the non-HR patients. Potentially the data from ANBL0531 could be used to show that less PR is an appropriate end point (not VGPR) in certain groups. Additionally, the one paper that was cited that addresses INSS stage 1-3 tumors could include a combination of high and non-high risk tumors, which could be difficult to make conclusions about.

Overall, the writing and presentation is very clear and a very useful summary of the surgical literature.

Author Response

This is a very nice review of the surgical considerations in neuroblastoma. The great majority of the paper seems to be dedicated to high risk disease, though not exclusively.

In some areas, such as the techniques for diagnosis, there is sufficient differentiation between non-HR and HR. In other areas, most prominently the section describing the studies looking at extent of resection, the discussion is very focused on HR disease without much acknowledgment of the lower risk tumors. It would be helpful to include some information about resection approaches in the non-HR patients. Potentially the data from ANBL0531 could be used to show that less PR is an appropriate end point (not VGPR) in certain groups. Additionally, the one paper that was cited that addresses INSS stage 1-3 tumors could include a combination of high and non-high risk tumors, which could be difficult to make conclusions about.

We thank the reviewer for this comment. The primary focus of this review was the surgical management of high-risk disease due to the challenges associated with the aggressive nature of high-risk neuroblastoma and its propensity to encapsulate vital structures. However, we agree that a brief overview of the available literature for low and intermediate risk disease would be of value. We have subsequently added several sentences to the first paragraph of the “Optimal Timing of Surgical Resection” section, as well as several more in-depth paragraphs to the beginning of the “Optimal Extent of Surgical Resection” section. These new paragraphs discuss the data regarding low and intermediate risk disease, including a discussion of ANBL0531. They are copied below:

“The goal of surgical management for low-risk neuroblastoma is complete resection of the primary tumor. Many patients have an overall survival approaching or exceeding 90% with surgical resection alone, particularly in the setting of asymptomatic, localized disease with favorable histology.4,19,33-36 In fact, asymptomatic infants with small adrenal tumors and those with stage MS disease without hepatomegaly can typically be offered a trial of observation alone.37,38Chemotherapy is typically reserved for: patients whose tumors cannot be resected without unacceptable morbidity (i.e., unresectability due to encapsulation of vital structures), patients who undergo a less than 50% resection of their primary tumor or who demonstrate progressive disease, and those presenting with life-threatening symptoms or neurologic compromise.19 However, these patients tend to demonstrate favorable outcomes even with less intense chemotherapy.19Disease recurrence is typically rare, and when it does occur, it is typically a locoregional recurrence that can be salvaged with surgery and/or chemotherapy.19

The optimal extent of surgical resection in intermediate-risk disease remains controversial.19 This may be due to the substantial heterogeneity of tumors included in this risk group. Patients with intermediate-risk neuroblastoma typically undergo biopsy followed by upfront chemotherapy to decrease tumor size, mitigate IDRFs and difficulty of resection, and/or to eliminate the need for resection.20 In COG trial ANBL0531 (n=404 patients with non-MYCN-amplified tumors), patients were treated with two, four, or eight cycles of upfront chemotherapy with or without surgery.20 Treatment group was assigned based on prognostic factors, including unfavorable histology, diploidy, or segmental chromosomal aberrations.20 Ultimate duration of therapy was determined by treatment response.20 Most patients demonstrated a partial response (PR, 50-90% reduction of the primary tumor) or better and therefore did not undergo surgical resection.20Surgical resection was utilized in patients to achieve the desired therapy goal without incurring substantial risk to vital structures, whereas additional chemotherapy was provided to patients who did not meet the therapy goal.20 ANBL0531 demonstrated that a reduction in therapy was appropriate for many patients with intermediate-risk disease, particularly those with localized disease, using an end-of-therapy goal of PR, as opposed to very good partial response (VGPR, greater than 90% reduction of the primary tumor).20” 

We have also added a statement in our discussion of the Fischer et al. (INSS stage 1-3 tumors) paper explicitly stating that it included patients from every risk group to prevent any confusion when drawing conclusions.

Overall, the writing and presentation is very clear and a very useful summary of the surgical literature.

                       We thank the reviewer for this gracious feedback.